# Assembly Characteristics and Influencing Factors of the Soil Microbial Community in the Typical Forest of Funiu Mountain

**DOI:** 10.3390/microorganisms12112355

**Published:** 2024-11-18

**Authors:** Kunrun He, Yiran Lai, Shurui Hu, Meiyi Song, Ye Su, Chunyang Li, Xinle Wu, Chunyue Zhang, Yuanhang Hua, Jinyong Huang, Shujuan Guo, Yadong Xu

**Affiliations:** 1School of Life Sciences, Zhengzhou University, Zhengzhou 450001, China; m18436625521@stu.zzu.edu.cn (K.H.); hushurui@stu.zzu.edu.cn (S.H.); smy1111@stu.zzu.edu.cn (M.S.); suye@stu.zzu.edu.cn (Y.S.); lcyangzzu@stu.zzu.edu.cn (C.L.); hjy666@zzu.edu.cn (J.H.); 2School of Agricultural Sciences, Zhengzhou University, Zhengzhou 450001, China; laiyiran0708@163.com (Y.L.); 18697726620@163.com (X.W.); c1278846310@163.com (C.Z.); yzchhyh@163.com (Y.H.); 3Henan Funiu Mountain Biological and Ecological Environment Observatory Research Project, Zhengzhou 450001, China

**Keywords:** litter, soil bacteria, environmental factors, community assembly, neutral theory

## Abstract

Assessing the relationship between litter characteristics and soil microbial community traits across different forest types can enhance our understanding of the synergistic interactions among litter, soil, and microorganisms. This study focused on three representative forest types in the Funiu Mountains—*Larix gmelinii* (LG), *Quercus aliena* var. *acutiserrata* (QA), and *Quercus aliena* var. *acutiserrata* + *Pinus armandii* (QAPA). The findings indicated no significant differences in Chao1 among the three forests; however, the Shannon index exhibited an initial increase followed by a decline. NMDS and ANOSIM analyses revealed significant structural differences across these forest types. Network topological metrics (nodes, edges, average degree, and average path distance) for bacterial taxa were higher in LG and QA compared with QAPA. Additionally, LG and QA demonstrated significantly greater average niche breadth than QAPA. The results from the null models (the proportion occupied by dispersal limitation is 62.2%, 82.2%, and 64.4% in LG, QA, and QAPA), modified stochasticity ratio (LG: 0.708, QA: 0.664, and QAPA: 0.801), and neutral community models (LG: R^2^ = 0.665, QA: R^2^ = 0.630, and QAPA: R^2^ = 0.665) suggested that stochastic processes predominantly govern the assembly of soil bacterial communities. Random forest analysis alongside Mantel tests highlighted LTP (litter total phosphorus), STN (soil total nitrogen), MCP (carbon-to-phosphorus ratio of microbial biomass), and SCN (soil carbon-to-nitrogen ratio) as critical factors affecting bacterial niche width; conversely LCN (litter carbon-to-nitrogen ratio), RCP (ratio of dissolved carbon to phosphorus), MCP, and SCN emerged as key determinants influencing community assembly processes. Furthermore, the PLS-SEM results underscored how both litter characteristics along with soil properties—and their associated alpha diversity—impact variations in niche breadth while also shaping community assembly dynamics overall. This research provides vital insights into understanding synergistic relationships between litter quality, soil characteristics, and microbial community across diverse forest ecosystems.

## 1. Introduction

Soil is a dynamic and multifaceted ecosystem that plays a critical role in sustaining life on Earth. It acts as a foundation for plant growth, a habitat for diverse microorganisms, and a key player in nutrient cycling and organic matter decomposition. Among the myriad factors influencing soil ecosystem properties, litter emerges as a pivotal element. Litter serves as a crucial mechanism for the reintroduction of organic matter and nutrients into the soil, constituting an essential component of terrestrial ecosystems [1,2]. The substrate and nutrients essential for plant growth are primarily derived from the soil. Throughout their development, these elements are predominantly returned to the soil in the form of litter. The rates of decomposition and nutrient release from litter play a critical role in determining the supply and equilibrium of nutrients within the soil, thereby influencing its physical, chemical, and biological properties [3]. As a crucial intermediary in plant–soil nutrient feedback mechanisms, litter can influence soil microbial communities both directly and indirectly through its chemical composition, which encompasses characteristics such as cellulose content, lignin concentration, and the carbon-to-nitrogen ratio [4,5,6]. Soil microorganisms decompose non-biological organic matter into simpler compounds, primarily to facilitate their growth, development, and reproduction. While extensive research has been conducted on the effects of litter on soil microbial diversity, composition, and biomass [7,8,9], there remains a paucity of studies investigating the relationship between litter properties and microbial community assembly; notably, the specific role of bacteria in this context remains poorly understood.

The investigation of microbial community assembly processes plays a crucial role in elucidating the underlying mechanisms governing microbial community diversity, composition distribution, and ecosystem functioning [10,11,12,13]. By comprehending the assembly process of microbial communities and their pivotal role in material cycling and energy flow within forest ecosystems, it also aids in optimizing soil management and forest management practices. Existing ecological theories propose that the assembly process of microbial communities is primarily influenced by two complementary mechanisms: deterministic processes and stochastic processes [10,14,15]. The former is grounded on niche theory, which governs community composition and species abundance by shaping specific niches for species with distinct characteristics, involving non-biotic environmental selection as well as the regulation of various biotic interactions such as predation, competition, mutualism, commensalism, and facilitation. This leads to substantial alterations in community structure, particularly under different environmental conditions where beta diversity demonstrates greater variation [12,16,17]. The latter is based on neutral theory, assuming a random equilibrium between species loss and gain; all species possess equal ecological niches. It predominantly encompasses birth and death rates, immigration and emigration events, along with speciation and extinction scenarios [18,19,20]. The research indicates that the relative contributions of deterministic processes and stochastic processes to the assembly process of microbial communities can be determined by environmental variables such as soil properties and vegetation types. For instance, Zhu et al. [21] discovered that elevated levels of available phosphorus and total phosphorus in soils from long-term fertilized farmland can enhance the stochastic processes of bacterial communities, whereas a higher soil carbon-to-phosphorus ratio and nitrogen-to-phosphorus ratio can suppress the stochastic processes of bacterial communities. Furthermore, there exists a positive correlation between bacterial β-diversity and the stochastic processes governing their communities. The differences in soil pH and C:N ratio are the primary driving factors for community succession between paired grassland and forest sites. The variation in pH within successional stages is a crucial factor leading to the relative dominance of deterministic processes, while abiotic factors play a selective role in shaping bacterial community composition [22]. Soil pH plays a differential role in driving bacterial community assembly processes across different ecosystems, such as forests [23,24], grasslands [25], and farmlands [26], and it significantly impacts both stochastic and deterministic processes. Changes in acidic, neutral, or alkaline pH environments alter the proportions of contributions from stochastic and deterministic processes. Research has indicated that changes in forest types can directly or indirectly regulate soil biota through litter decomposition, thereby affecting soil ecological processes [2,3,27]. Different types of litter provide varying resources and habitat environments, which can significantly alter soil stoichiometry, pH, nutrient availability, and organic matter content. These changes ultimately affect the reproduction and survival environment of specific microorganisms, thereby influencing the spatial distribution and community composition of these organisms [28,29,30]. Hence, we deduce that the composition of litter exerts an influence on the interplay between stochastic and deterministic processes in shaping bacterial communities within forest ecosystems. 

Funiu Mountain is situated in the central-western region of Henan Province, within the heartland of the Central Plains. It falls within a transitional zone between subtropical and warm temperate climates and lies on the boundary line separating northern and southern China. Renowned for its vast expanse of natural forests and exceptional biodiversity, it boasts the largest area of such forests in Henan Province. The predominant forest types found here encompass *Larix gmelinii* (LG), *Quercus aliena* var. *acuteserrata* (QA), and the mixed forest of *Quercus aliena* var. *acutiserrata* and *Pinus armandii* (QAPA). These forests present an excellent opportunity to investigate the relationship between litterfall and soil microbial community assembly processes across different forest types. This study aims to elucidate variations in litterfall properties, soil characteristics, and microbial community diversity and composition between distinct forest ecosystems (LG, QA, and QAPA), as well as changes in co-occurrence networks among microorganisms, while examining niche breadth and community assembly processes. This study primarily seeks to address the following questions: (i) To what extent do litter traits, soil characteristics, and bacterial community features vary between different forest ecosystems? (ii) What are the assembly processes and niche breadths of bacterial communities across these diverse forest ecosystems? (iii) How do the litter and soil components jointly influence the assembly process of the community? 

## 2. Materials and Methods

### 2.1. Study Area

The study area is situated within the Funiu Mountain National Nature Reserve (33°40′ N~34°20′ N, 110°0′ E~112°45′ E), which falls within the transitional region between the North Subtropical and Warm Temperate zones. It exhibits a continental monsoon climate with an average elevation ranging from 750 m to 1450 m. The annual mean temperature ranges from 12.1 °C to 15.1 °C, while the annual average precipitation varies between 800 mm and 1100 mm, primarily concentrated from April to September, accounting for approximately 70% of the total rainfall. The soil type prevalent in this area is yellow-brown soil. Moreover, the Funiu Mountain region serves as a watershed for three major river systems—the Yangtze River, Yellow River, and Huai River—along with their tributaries. This region boasts intricate topography, a high forest coverage rate, a substantial timber stock volume, and abundant biodiversity that collectively contribute to its relatively stable ecosystem dynamics. Notable tree species encompass *Larix gmelinii*, *Quercus variabilis*, *Quercus aliena* var. *acuteserrata*, and *Pinus armandi*, among others, whereas dominant herbaceous communities include *Carex lanceolata*, *Melampyrum roseum*, *Cardamine macrophylla*, *Carex siderosticta*, and *Rodgersia aesculifolia*. 

### 2.2. Litter and Soil Sampling

Three representative forest types (*Larix gmelinii*, *Quercus aliena* var. *acuteserrata*, and *Quercus aliena* var. *acutiserrata* + *Pinus armandii*) were selected, and five 10 × 10 m quadrats were established in July 2023. The basic characteristics of the experimental site are provided in Appendix A and Figure 1. Five cores (4 cm in diameter, 20 cm in height) were randomly obtained from each plot and thoroughly mixed to create a composite soil sample and repeated twice in each quadrat. Simultaneously, the litter from the surface of each soil core was collected. Consequently, 30 soil and 30 litter samples (10 replicates) were obtained. A portion of each soil sample was immediately shipped from the field to the laboratory in an ice box and immediately stored at −80 °C for DNA extraction. The remaining portion of soil samples was air-dried at room temperature and stored for physicochemical analysis.

### 2.3. Determination of Litter and Soil Properties

The acid detergent fiber (ADF) method was used to determine litter cellulose (LC) and lignin (LL) content [31]. A potassium dichromate volumetric method was used to determine litter organic carbon (LOC) and soil organic carbon (SOC). The Kjeldahl method was used to determine litter total nitrogen (LTN) and soil total nitrogen (STN) [32]. Litter total phosphorus (LTP) and soil total phosphorus (STP) were determined using the molybdenum–antimony resistance colorimetric method. The soil water content (SWC) was measured gravimetrically after drying at 105 °C for 24 h. The soil pH was measured using a pH meter. The SBD was calculated using the ring knife method. A continuous flow analyzer (AutoAnalyzer-AA3, Seal Analytical, Norderstedt, Germany) was used to measure the soil ammonium nitrogen (AN) and nitrate nitrogen (NN) [32]. The soil available phosphorus (AP) was determined by NaHCO3 extraction and the molybdenum–antimony resistance colorimetric method. Soil microbial biomass C (MBC), N (MBN), and P (MBP) were measured using the chloroform fumigation-extraction method [33]. The concentrations of dissolved organic carbon (DOC) and dissolved organic nitrogen (DON) in unfumigated soil samples were quantified using MBC and MBN analysis [34]. The litter and soil properties in different forests are provided in Appendix A.

### 2.4. Extraction of Soil DNA and Sequencing Analysis of the 16S rRNA Gene

Total DNA was extracted from soil samples using the E.Z.N.A.^®^ Soil DNA Kit (Omega Biotek, Norcross, GA, USA). The extracted DNA samples were then sent to Shanghai BIOZERON Co., Ltd. (Shanghai, China) for the sequencing of the *16S rRNA* gene on an Illumina MiSeq platform (Illumina, San Diego, CA, USA). PCR amplification of the V3-V4 regions of bacterial 16S rRNA genes was performed using the primers 338F (5′-ACTCCTACGGGAGGCAGCA-3′) and 806R (5′-GGACTACHVGGGTWTCTAAT-3′). Each sample’s PCR products were evaluated through electrophoresis on a 2.0% agarose gel and subsequently purified with the AxyPrep DNA Gel Extraction Kit (Axygen Biosciences, Union City, CA, USA). The purified PCR products were quantified using a Qubit^®^ 3.0 (Life Invitrogen, Norcross, GA, USA). These pooled products were then utilized to construct a DNA library. The raw sequence data presented in this study have been archived in the Genome Sequence Archive in the National Genomics Data Center (Nucleic Acids Res 2024), China National Center for Bioinformation/Beijing Institute of Genomics, Chinese Academy of Sciences (GSA: CRA 030759) and are publicly accessible at https://ngdc.cncb.ac.cn/gsa (accessed on 14 October 2024).

### 2.5. Bioinformatics Analysis

The raw FASTQ data were demultiplexed using custom Perl scripts based on the barcode sequence information for each sample within QIIME2. The DADA2 plugin in the QIIME2 2023.5 software was then employed to process the aired reads, eliminate chimeras, and cluster the clean data into amplicon sequence variants (ASVs) [35]. Each ASV was assigned to a taxon on the basis of the SILVA database (Release138.1 http://www.arb-silva.de) (accessed on 20 August 2023) [36]. The alpha diversity indices of the bacterial communities, such as the Chao1 and Shannon index, were calculated. Nonmetric multidimensional scaling (NMDS) was performed to assess the structural variations in the soil microbial community using the Bray–Curtis distance (first two axes) at the ASV level followed by an analysis of similarities (ANOSIM) to evaluate the significance of forest types on microbial community composition. The niche breadth of the bacterial communities was measured using the ‘spaa’ (version 0.2.2) package in R v4.0.5. The null model was used to identify the process governing microbial community assembly based on the βNTI and Raup–Crick metric (RC), which was carried out using the “iCAMP” (version 1.3.4) and “ape” (version 5.7-1) packages [37,38]. The modified stochasticity ratio (MST), which was conducted by the “NST” (version 3.1.10) package, was used to further evaluate bacterial community assembly. The Sloan neutral model was employed to estimate the relative contribution of stochastic processes to the assembly of microbial communities, and the model fitting was performed as previously described [39]. Microbial co-occurrence networks were constructed on the basis of the random matrix theory (RMT) method with the same threshold [40]. Networks were graphed in Gephi. Additionally, two parameters, within-module degree (Zi) and among-module connectivity (Pi), were further used to assess the topological roles of different ASVs. The nodes were classified into four categories, including peripherals, module hubs, network hubs, and connectors, as previously described [41,42].

### 2.6. Statistical Analyses

One-way analysis of variance (ANOVA) was used to evaluate differences in properties between samples. Tukey’s HSD test was performed for multiple comparisons. To alleviate multicollinearity and optimize the number of litter and soil factors, we initially conducted a screening of the collected variables through stepwise regression. Concurrently, we applied a variance inflation factor (VIF) threshold of less than 5 to identify and mitigate issues related to multicollinearity. Spearman’s correlation analysis was employed to explore the associations between soil properties and microbial community diversity and structure. The Mantel test was used to analyze the relationship between alpha diversity, community composition, soil characteristics, and microbial diversity and their interaction patterns. Random forest was performed to determine the role of litter and soil properties in the niche breadth and assembly of the microbial community. Partial least squares structural equation modeling (PLS-SEM) was applied to quantify the direct and indirect effects of litter and soil variables on niche breadth and community assembly. We employed goodness of fit (GOF) as a metric to assess the overall quality of our model, categorizing its fit into weak, moderate, and strong levels on the basis of GOF cutoff values of 0.1, 0.25, and 0.25, respectively. Statistical significance was set at *p* < 0.05. Spearman’s correlations and the Mantel test were visualized using R software via the “linkET” package (version 0.0.2.4). The “plspm” (version 0.5.1) package was utilized for PLS-SEM modeling. The null model served to assess the assembly process within the contributing soil microbial community. Box plots, scatter plots, and bar graphs were plotted using Origin 2019 (Origin Lab Corporation, Northampton, MA, USA).

## 3. Results

### 3.1. Soil Bacteria Diversity and Community Composition

The results presented in Figure 2 indicate that there was no significant difference in Chao1 among different forest ecosystems (Figure 2a), while the Shannon index demonstrates an increasing trend followed by a decreasing trend (Figure 2b). Furthermore, the NMDS and ANOSIM analyses revealed significant differences in the soil microbial community structure between different forest ecosystems (Figure 2c). At the phylum level, Acidobacteriota, Proteobacteria, Verrucomicrobiota, and Actinobacteriota were identified as the main dominant phyla, collectively representing over 80% of the relative abundance on average (Figure 2d).

### 3.2. Co-Occurrence Network Structure and the Keystones

We constructed microbial co-occurrence networks to explore microbial associations and found that with the same similarity thresholds, the network had different properties (Figure 3a–c and Appendix A). The network topological characteristics (nodes, edges, average degree, and average path distance) of the bacteria taxa in LG and QA were higher than those in QAPA. All three forests had high modularity, which was higher in QAPA. The proportion of positive correlations exhibited a trend of first decreasing and then increasing, while the proportion of negative correlations exhibited the opposite trend. We identified a series of module hubs and connectors hubs based on their within-module connectivity (Zi) and among-module connectivity (Pi), which could be regarded as keystones that play key roles in shaping network structure (Figure 3d). The number of module hubs and connectors hubs was higher in LG (n = 6) and QA (n = 1) than in QAPA (n = 0). The keystones mainly belonged to Acidobacteria, Verrucomicrobia, and WPS−2.

### 3.3. Soil Bacteria Niche Breadth and Community Assembly Process

The average niche breadth of LG and QA was significantly greater than that of QAPA (Figure 4a). The null model revealed that dispersal limitation and heterogeneous selection were the main ecology processes, and the proportions occupied by dispersal limitation were 62.2%, 82.2%, and 64.4% in LG, QA, and QAPA, respectively (Figure 4b). Using MST quantification, the relative importance of deterministic processes and stochastic processes in bacterial communities was determined. For the bacterial communities, the average MST values between the three forest ecosystems were 0.708, 0.664, and 0.801, respectively, all exceeding the threshold line of 0.5 (Figure 4c). We also used the neutral community model (NCM) to successfully estimate a large fraction of the relationship between the occurrence frequency of OTUs and their relative abundance variations (Figure 4d–f), with 66.5%, 63.0%, and 66.5% of explained community variance, and the m values were estimated to be 0.0104, 0.0101, and 0.0127 for LG, QA, and QAPA, respectively.

### 3.4. Correlations Between Litter Properties, Soil Characteristics, and Microbial Community

Correlation analyses revealed that the Shannon index was only negatively correlated with SCN. In addition, no litter and soil factors were correlated with Chao1. Low-abundance phyla, such as Methylomirabilota, Gemmatimonadota, and Myxococcota, displayed varying degrees of negative correlation with litter properties, as well as negative correlation with STN, SCN, and MCP, while showing a positive correlation with SWC. High-abundance dominant phyla, such as the Acidobacteriota phylum, exhibited a significant negative correlation with SWC and RCP, along with a significant positive correlation with AP. Actinobacteriota demonstrated a significant positive correlation with LL, LTP, and SCN while displaying a significant negative correlation with SWC. Proteobacteria showed a notable positive correlation with RCP and a marked negative correlation with AP. Chloroflexi displayed substantial negative correlations with LC, LON, and RCP (Figure 5a). As the environmental distance increased, there was a significant decrease in the similarity of microbial communities (Figure 5b). Random forest showed that STN (5.69%), MCP (7.64%), and SCN (7.70%) were the key predictors of soil bacteria niche breadth (Figure 5c). Moreover, MCP (5.56%) and SCN (7.62%) were crucial predictors of soil bacteria MST (Figure 5d). Mantel test results showed that alpha diversity was related to STN and AN, and community composition was connected to LC. LTP (r: 0.223, p: 0.047) and SCN (r: 0.242, p: 0.005) were crucial factors affecting the niche breadth of bacteria. LCN (r: 0.244, p: 0.030) and RCP (r: 0.157, p: 0.049) were important factors affecting bacteria community assembly (Figure 5e).

### 3.5. Direct and Indirect Effects of Litter, Soil Properties, Alpha Diversity, and Community Composition on Niche Breadth and Community Assembly

PLS-SEM explained 34.5%, 13.0%, 5.6%, 40.7%, and 85.1% of the variation in soil properties, alpha diversity, community composition, community assembly, and niche breadth, respectively, with an overall GOF of the PLS-SEM of 0.423 (Figure 6a). The increase in niche breadth was substantially influenced by alpha diversity, with a standardized effect of 0.722, followed by soil properties, with a direct standardized effect of −0.314 and a total standardized effect of −0.571. The total effect of litter on the niche breadth was primarily mediated by its significant influence on soil properties, alpha diversity, and community composition, with a significant indirect standardized effect of −0.386 (Figure 6b). No significant direct standardized effects of factors on the community assembly process were observed. Notably, soil properties had significant total effects on the community assembly process (standardized effects = 0.566). The total positive effect (standardized effects = 0.365) of litter on the community assembly process was primarily mediated by its significant influence on soil properties.

## 4. Discussion

### 4.1. Variation in Soil Microbial Diversity and Community Composition Across Different Forest Types

Chao1 is a widely used indicator for estimating species richness, primarily reflecting the number of species in a given sample [43]. Despite variations observed between forest types, these differences did not exert a significant influence on the species richness of soil microorganisms (Figure 2a). Furthermore, correlation analysis revealed no significant relationship between Chao1 and soil or litter factors (Figure 5a). These findings suggest that the region provides comparable microbial habitats for diverse forest ecosystems, potentially contributing to stability in terms of species abundance within soil microbial communities. This phenomenon may be attributed to the adaptability of microorganisms to environmental conditions at various regional scales [44,45]. However, the Shannon index results exhibited a pattern of initial increase followed by a subsequent decrease (Figure 2b), indicating temporal fluctuations in species diversity across diverse forest ecosystems. The Shannon index not only considers species richness but also accounts for their evenness [46]. Hence, alterations in this index may suggest that Quercus aliena var. acutiserrata (QA) forests possess a higher degree of evenness in microbial community structure, facilitating enhanced survival and competition among microbial species. Nevertheless, under changing environmental conditions or resource allocation dynamics, certain specific species might dominate the community, leading to an overall decline in diversity. This trend could potentially be attributed to variations in soil fertility, stoichiometry ratio, vegetation type, and other environmental factors within forest ecosystems [47]. The correlation analysis findings (Figure 5a) reveal a significant negative association between SCN and the Shannon index. This discovery implies that organic matter quality significantly influences microbial diversity within the soil system. SCN serves as an important indicator for assessing organic matter decomposition rate and microbial growth in soils [48,49,50]. A low SCN typically indicates nitrogen-rich soil, which promotes microbial reproduction and diversity; conversely, a high SCN may hinder microbial growth, resulting in decreased diversity.

The results of NMDS and ANOSIM revealed significant disparities in the soil microbial community structure between distinct forest ecosystems (Figure 2c). This suggests that while species richness does not exhibit notable differences, there are substantial variations in the composition of microbial communities between different forest types. This phenomenon is likely closely associated with dissimilarities in soil physicochemical properties, such as pH value, nutrient availability, and stoichiometric ratios, across diverse forest types [44,48]. Divergent environmental conditions may facilitate the adaptation of specific microbial communities to particular ecosystem conditions, leading to pronounced dissimilarities in community structure. Further analysis at the phylum level demonstrated that Acidobacteriota, Proteobacteria, Verrucomicrobiota, and Actinobacteriota were dominant phyla with a relative abundance exceeding 80% (Figure 2d). These phyla typically play crucial ecological roles within soil microbial communities [51,52]. The phyla Acidobacteria and Bacteroidetes are commonly involved in the degradation of organic matter and nitrogen cycling, while Actinobacteria is renowned for its potent decomposition abilities, playing a pivotal role in the processes of organic matter decomposition and nutrient cycling. The contribution of Verrucomicrobiota has been relatively understudied but may potentially contribute to soil health under specific conditions [53]. The Mantel test results also demonstrated significant correlations between alpha diversity and STN as well as AN levels in the soil, whereas community composition exhibited significant associations with LC content. Therefore, the decomposition of litter materials and nutrient cycling within forest ecosystems exhibit intricate relationships with microbial diversity and community composition [2].

### 4.2. Soil Co-Occurrence Networks and Key Species in Different Forest Types

Firstly, the topological characteristics of the networks in LG and QA were higher than those in QAPA, suggesting that in these two types of forests, the complexity of microbial communities was greater, and the potential interaction relationships were also more intimate. A high average degree and a short average path distance imply that these communities possess higher efficiency and stability, with faster resource exchange among microorganisms [54]. Although there are fewer direct connections, information or influences in the network can spread relatively rapidly among microorganisms, and the response capacity of the ecosystem is stronger. By contrast, the microbial network complexity in QAPA was lower, which might be related to the complication of resource allocation resulting from the coexistence of multiple tree species in the mixed forests. Tree species diversity may weaken the competitiveness of certain dominant microbial groups and reduce the complexity of their networks [55]. Additionally, this study discovered that the networks of all three forest types exhibited a high level of modularity, indicating that the functions of microbial communities were clearly divided. Different modules might represent adaptation strategies to different environmental conditions or distinct ecological functions. This modular characteristic is conducive to maintaining the stability of the community because when a certain module is disrupted, other modules can maintain normal functions, thereby enhancing the system’s anti-interference ability. Notably, the modular level in the mixed forests of QAPA was relatively high, which might be related to tree species diversity. More tree species provide a more heterogeneous habitat for microorganisms and promote the differentiation of different microbial groups. Furthermore, the high modularity of QAPA may also be due to the significant improvement in soil nutrient availability brought about by the mixed forest, which promotes the differentiation of soil bacterial ecological niches [56]. The increase in ecological niche differentiation may lead to a decrease in the number of microbial interactions between bacteria [57], resulting in fewer connections between bacteria. Finally, through the analysis of within-module connectivity (Zi) and between-module connectivity (Pi), the researchers identified key module hubs and connector hubs. The number of hubs in LG and QA was much higher than that in QAPA, indicating that there were more key microbial groups in these forests dominated by a single tree species, which might play a core role in maintaining the structure and function of the community. The discovery of Acidobacteria, Verrucomicrobia, and WPS-2 as key bacteria phyla is in line with the conclusions in many studies. These bacterial groups usually play significant roles in nutrient cycling and organic matter degradation. This finding also confirms that certain low-abundance microorganisms actively contribute to driving material cycling and shaping the functional composition of the community, thereby increasing their likelihood of interaction with other microbial communities, which aligns with the results of Mary et al. [58] and Zhang et al. [59].

### 4.3. Soil Bacterial Niche Breadth and Community Assembly Processes in Diverse Forest Ecosystems

The niche breadth of bacteria pertains to the gamut of resources they exploit within the milieu; a more expansive niche breadth typically implies that bacterial communities can avail themselves of a more extensive array of resources, thereby fostering higher community diversity [60]. Our observations reveal that the niche breadth of bacteria in the soils of QA and LG forests is significantly greater than that detected in the soils of QAPA mixed forests (Figure 4a). This revelation intimates that disparate forest types, concomitant with variations in litter and soil environments, exert disparate impacts on the niche breadth of bacterial communities [56]. In mixed forests, the augmented plant diversity and biomass contribute to a more intricate and heterogeneous soil resource panorama. Microorganisms dwelling in such environments may adopt more collaborative or competitive tactics to negotiate these complex resource dynamics, potentially giving rise to contracted niche breadth [61]. Moreover, the composition and quantity of root exudates from diverse tree species may mold the structure of soil microbial communities. As a consequence, microorganisms might constitute specific communities tailored to these plant-derived resources, which could also engender narrower niche breadth—aligning with the witnessed trends in diversity outcomes. Additionally, the outcomes from the structural equation modeling further illuminate that alpha diversity significantly influences the niche breadth (Figure 6a).

The findings from the null model suggest that both diffusion limitation (a stochastic process) and heterogeneous selection (a deterministic process) collaboratively influence the assembly of bacterial communities. Specifically, the contributions of diffusion limitation in LG, QA, and QAPA were 62.2%, 82.2%, and 64.4%, respectively (Figure 4b). The dominance of diffusion limitation within these forest ecosystems indicates that the distribution of microbial communities is constrained by their spatial migration capabilities. This relatively high proportion of diffusion limitation may reflect a pronounced geographical dependence on the spatial distribution patterns of microorganisms within forest soils. In environments characterized by greater diffusion limitations, the spatial heterogeneity among microbial communities is likely to be more pronounced [62]. The quantification of ecological processes through the modified stochasticity ratio (MST) indicates that the average MST values for bacterial communities across the three forest ecosystems exceed 0.5, specifically recorded at 0.708, 0.664, and 0.801, respectively (Figure 3c). This finding suggests a predominant role of stochastic processes in shaping these bacterial communities. The neutral community model (NCM) effectively estimated the relationship between the occurrence frequencies of most amplicon sequence variants (ASVs) and variations in their relative abundances (Figure 4d–f), accounting for community variances of 66.5% and 63.0% and another instance at 66.5%. The estimated m values were determined to be 0.0104, 0.0101, and 0.0127, indicating that neutral processes—such as random migration and extinction—significantly influence community structure within these forest ecosystems. Furthermore, the high explanatory power of the NCM underscores its applicability across diverse forest environments and its capacity to elucidate both structural patterns and dynamic interactions within bacterial communities [63]. A smaller m value indicates that non-stochastic processes, such as environmental selection and intraspecific competition, also exert significant influences on community structure. This finding may be linked to the high mobility and extinction rates of microorganisms within forest ecosystems, which contribute to a pronounced randomness in bacterial communities across both temporal and spatial scales. All three assessment methods reveal that stochastic processes significantly shape the structure of bacterial communities. Specifically, under identical environmental conditions, the composition and structure of these communities can exhibit substantial variability and unpredictability. While deterministic processes (such as niche selection) may influence community structure under certain circumstances, the predominance of stochastic processes suggests that variations within these communities are more likely driven by stochastic factors. Previous studies have highlighted the importance of stochastic processes in the formation of microbial communities, which may be linked to the spatial variability of forest soil characteristics, neutral ecological dynamics, diverse species pools, dispersal mechanisms, and interactions among microbes [64,65]. Together, these factors create conditions where community composition appears to be largely influenced by random events rather than specific environmental selection pressures. However, it is important to recognize that some research has reached different conclusions [66]; this difference might arise from bacteria’s generally shorter generation times and higher transmission rates, allowing them to respond more quickly to changes in their environment and making them more susceptible to deterministic processes like environmental filtering or biological interactions. 

Additionally, the structural equation modeling results indicate that soil properties have a significant overall impact on the assembly of bacterial communities. Meanwhile, the findings from random forest and the Mantel test suggest that various factors influence the assembly of soil bacterial communities, including MCP, SCN (Figure 5d), LCN, and RCP (Figure 5e). The MCP provides insight into how microorganisms demand and utilize nutrients efficiently. Typically, a high MCP signals an abundance of carbon sources alongside phosphorus limitations in the soil; this can lead to certain microbial populations becoming dominant, thereby influencing community composition and structure [33]. Litter serves as a vital carbon source for soil microorganisms, while the LCN directly impacts organic matter decomposition rates. This ratio determines the energy and nutrients accessible to microorganisms during decomposition; specific types of organic matter decomposed by particular microorganisms may result in varied community structures [2]. The RCP reflects nutrient balance status, which plays a crucial role in microorganism growth and activity [33]. When nutrient supply is unevenly distributed, certain bacteria might gain competitive advantages that affect overall community diversity and stability. Additionally, SCN is commonly used as an indicator of soil fertility affecting both microbial growth and community structure. In soils with low SCN, active organic matter decomposition leads to heightened microbial activity [67]; thus, allowing microorganisms to access more nutrients promotes their biomass increase and fosters more complex community structures. In summary, the dynamic nature of microbial community assembly shifts according to environmental factor influences’ strength—resulting from intricate interactions among multiple factors.

## 5. Conclusions

This study elucidates the diversity, composition, niche breadth, and assembly processes of soil bacterial communities across various forest types in the Funiu Mountains. It further investigates the influences of litter and soil properties on these communities through correlation analysis, random forest, the Mantel test, and PLS-SEM. The findings underscore the significance of dispersal limitation and heterogeneous selection in community assembly while highlighting the predominance of stochastic processes within community structure. Support for this perspective is provided by both neutral community models and randomization correction approaches. Our results indicate that MCP and SCN are critical determinants influencing bacterial community assembly, with litter quality exerting an indirect effect on this process via alterations in soil properties. These insights offer a novel perspective for ecological research on forest soil microbial communities and provide theoretical foundations for future soil management practices and ecological restoration efforts. Meanwhile, we will establish long-term monitoring sites to regularly collect samples and analyze the trends of microbial communities, which will help predict the potential impact of future environmental changes on the health and function of forest ecosystems.

## Figures and Tables

**Figure 1 microorganisms-12-02355-f001:**
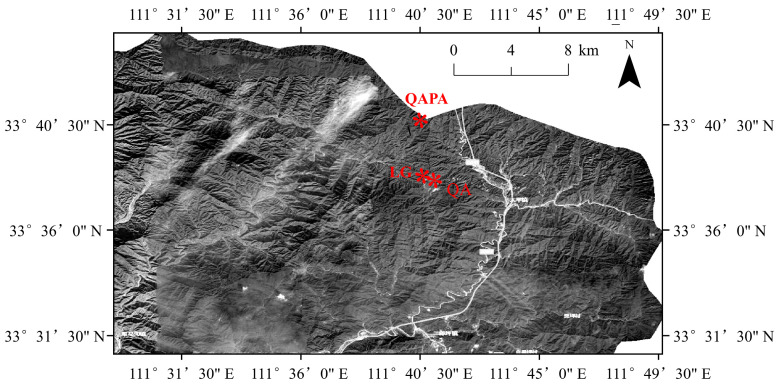
Geographic location of the study sites.

**Figure 2 microorganisms-12-02355-f002:**
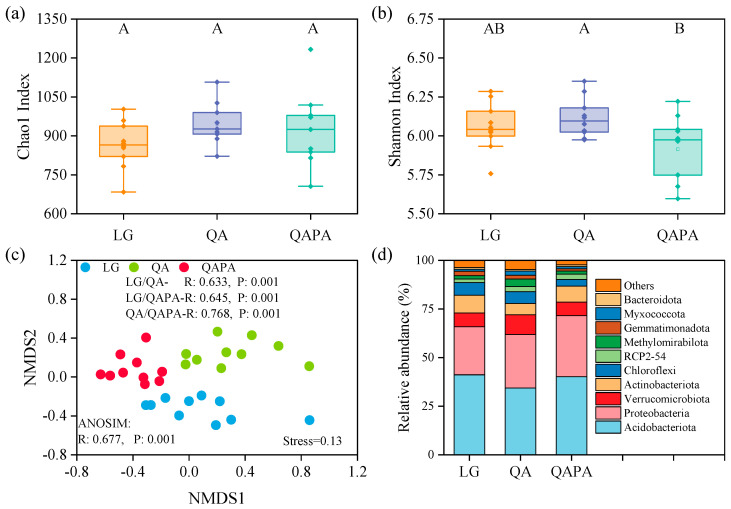
Soil diversity and community composition of different forest types. (**a**): Chao1; (**b**): Shannon index; (**c**): NMDS analysis. The distance between the samples represents the degree of difference, and the samples from the same group are represented by the same color. (**d**): Relative abundance of community at the phylum level. Different uppercase letters indicate significant differences among the forest soils (*p* < 0.05), and the same applies below.

**Figure 3 microorganisms-12-02355-f003:**
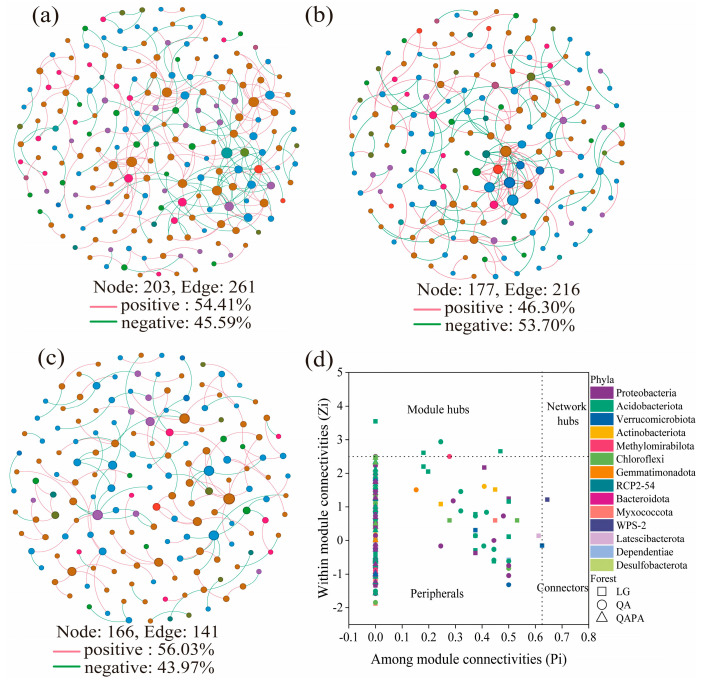
Co−occurrence networks and of the soil bacterial community in LG (**a**), QA (**b**), and QAPA (**c**), Zi–Pi plots showing the keystone species of bacteria (**d**) in different forests. The nodes represent the amplicon sequence variants (ASVs), and the size of each node is proportional to the degree of the ASVs.

**Figure 4 microorganisms-12-02355-f004:**
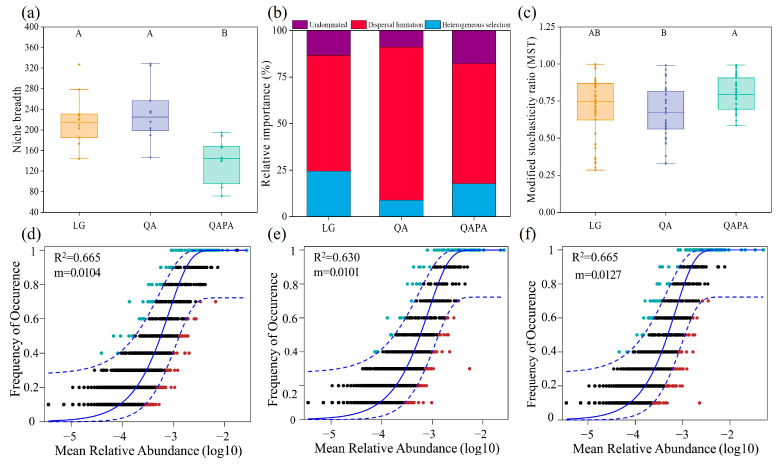
Niche breadth (**a**), null model (**b**), modified stochasticity ratio (**c**), and neutral community model ((**d**) LG; (**e**) QA; (**f**) QAPA) of soil bacteria in different forest types. The solid blue line represents the best-fit values of the neutral community model, the dashed blue line represents the 95% confidence interval of the model (estimated through 999 bootstraps), and OTUs with occurrence frequencies higher or lower than predicted by the neutral community model are displayed in different colors.

**Figure 5 microorganisms-12-02355-f005:**
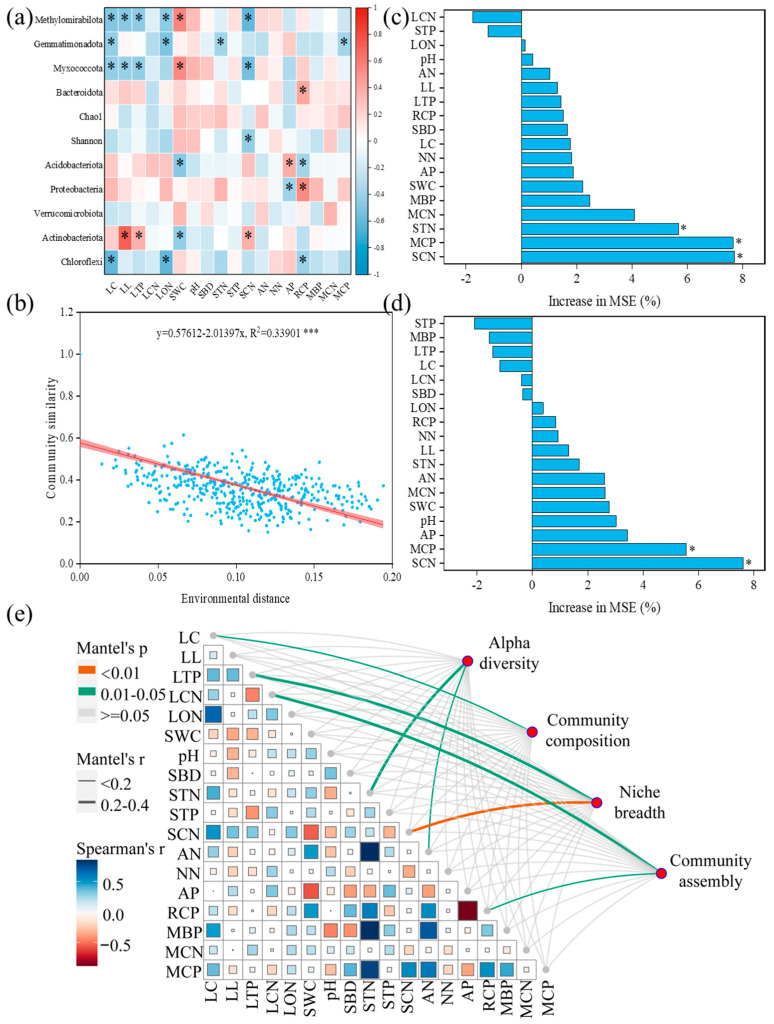
Correlations among soil bacteria diversity, community composition, and litter, soil properties (**a**). Relationships between the soil microbial community’s Bray–Curtis dissimilarities and the environmental distance (**b**). Random forest determined the role of litter and soil properties in the niche breadth (**c**) and the assembly (**d**) of bacteria. The Mantel test analyzed the relationship between litter, soil properties and alpha diversity, community composition, niche breadth, and assembly process of bacterial (**e**). * *p* < 0.05, *** *p* < 0.001.

**Figure 6 microorganisms-12-02355-f006:**
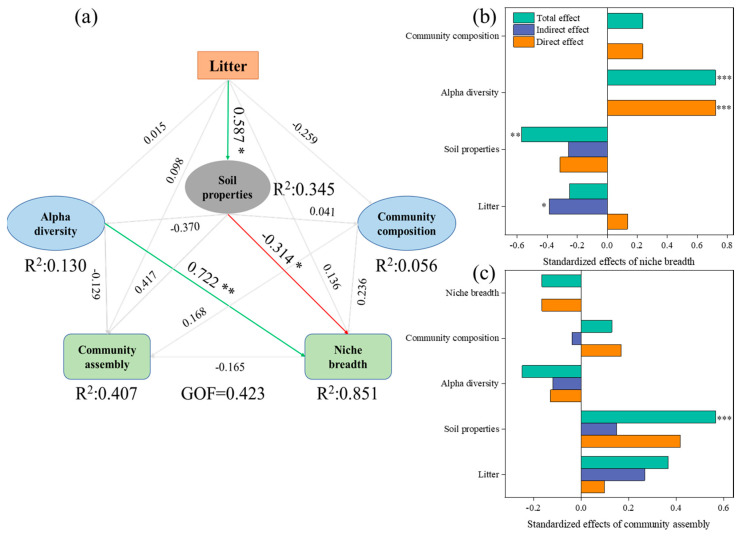
Effects of litter and soil properties on niche breadth and community assembly (**a**). The adjusted (adj.) R^2^ of the averaged model and the *p*-value of each variable are given as * *p* < 0.05, ** *p* < 0.01, and *** *p* < 0.001; red line: negative effects; green line: positive effects. The standardized effects of SEM ((**b**) niche breadth; (**c**) community assembly).

## Data Availability

The original data generated in this study are included in this article. Further inquiries can be directed to the corresponding author.

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
