# Peer review of "Assembly Characteristics and Influencing Factors of the Soil Microbial Community in the Typical Forest of Funiu Mountain"

_microorganisms, 2024, doi:10.3390/microorganisms12112355_

Round 1
Reviewer 1 Report
Comments and Suggestions for Authors
Dear Editor-in-Chief,
Thank you very much for making this very well-written work available.
The text has scientific merit and relevant information.
Below are some observations:
Line 22: Is this name correct, "Chao1 index"?
Line 29: Please add percentage values ​​of the results.
Line 119: What was the motivation or hypothesis of this study? This point is not clear in the text.
Line 145: It is "Litter" and not "litter".
Line 159: Add a citation to the method used.
Line 237: The nomenclature is different from that presented in the Abstract.
Line 246: The symbol for p is different, is it p or P?
Line 264: The font is too small to read, please improve it.
Line 278: This type of presentation is appreciated.
Line 282: The font size is too small to read, please improve it.
Line 300-302: How much were these results? How many %?
Line 305: Where is the description of Figure b?
Line 418: The citations are written incorrectly.
There is no comma between the author and the "et".
Author Response
Dear Editor-in-Chief,
Thank you very much for making this very well-written work available.
The text has scientific merit and relevant information.
Response: I sincerely appreciate your affirmation and encouragement regarding my article; your evaluation holds significant value for me. It is gratifying to learn that you consider this article scientifically valuable and rich in relevant, important information. Your recognition not only acknowledges my individual efforts but also validates the hard work of our entire team. Furthermore, I will carefully examine your invaluable suggestions as I strive to further optimize and enhance this paper in my ongoing research.
Below are some observations:
Comments1: Line 22: Is this name correct, "Chao1 index"?
Response1: Thank you very much for your suggestion. We have corrected the incorrect expression and have made changes throughout the entire paper.
Comments2: Line 29: Please add percentage values ​​of the results.
Response2: Thank you very much for your valuable advice. We have added the percentage values to make the results clearer.
Comments3: Line 119: What was the motivation or hypothesis of this study? This point is not clear in the text.
Response3: Thank you very much for your valuable advice. I apologize for not clearly presenting our hypothesis. I have now added them, specifically as follows:
This study primarily seeks to address the following questions: (i) To what extent do lit-ter traits, soil characteristics, and bacterial community features vary among different forest ecosystems? (ii) What are the assembly processes and niche breadths of bacterial communities across these diverse forest ecosystems? (iii) How do the litter and soil components jointly influence the assembly process of the community?
Comments4: Line 145: It is "Litter" and not "litter".
Response4: I apologize for the error and thank you for bringing it to my attention. It has been corrected.
Comments5: Line 159: Add a citation to the method used.
Response5: I apologize for neglecting to cite the literature on the methods used to measure these indicators. The citation has been supplemented.
Comments6:Line 237: The nomenclature is different from that presented in the Abstract.
Response6: Thank you for pointing out the incorrect notation. I have now changed the “Chao index” to “Chao1”.
Comments7: Line 246: The symbol for p is different, is it p or P?
Response7: I apologize for using an inappropriate term, and I have agreed to change it to P.
Comments8: Line 264: The font is too small to read, please improve it.
Response8: Thank you very much for your valuable advice. I have redrawn the figure and increased the font size.
Comments9: Line 278: This type of presentation is appreciated.
Response9: Thank you very much for your recognition.
Comments10: Line 282: The font size is too small to read, please improve it.
Response10: Thank you very much for your valuable advice. I have redrawn the figure and increased the font size.
Comments11: Line 300-302: How much were these results? How many %?
Response11: Thank you very much for your good advice. I have already added specific numerical values for the results.
Comments12: Line 305: Where is the description of Figure b?
Response12: I apologize for missing the annotation for Figure 4b. The error has been corrected.
Comments13: Line 418: The citations are written incorrectly.
There is no comma between the author and the "et".
Response13: Thank you very much for your careful review and valuable suggestions. The commas have been removed.
Reviewer 2 Report
Comments and Suggestions for Authors
The topic of manuscript is interesting and important from point of view relationship between litterfall and soil microbial community in different forest types. Collected samples have been processed by the statistical software, results are described well, discussion is very well written. Accept after minor revision.
Main comments:
Abstract is too long and must be shortened, Instructions for authors says that ″The abstract should be a total of about 200 words maximum″.
The map of the study areas should be added to the manuscript
Line 145 - ″litter″ should be ″Litter″
Lines 146-147, 345, 642 – the latin name of tree species should be in italic
Line 323 – ″asssembly″ should be ″assembly″
Could you include to the end of conclusion 1-2 sentences on plans for further research?
Chapter References must be formatted in accordance with the instructions for authors,
for example Journal Articles:
1. Author 1, A.B.; Author 2, C.D. Title of the article. Abbreviated Journal Name Year, Volume, page range.
Lines 536, 604 – CO2 - ″2″ should be as subindex
Line 630 – ″Abbreviated Journal Name Year, Volume, page range″ are missing
Author Response
The topic of manuscript is interesting and important from point of view relationship between litterfall and soil microbial community in different forest types. Collected samples have been processed by the statistical software, results are described well, discussion is very well written. Accept after minor revision.
Response: Thank you very much for your careful review of our manuscript and valuable comments. We are very pleased to see that you think our research topic is of great significance, especially in exploring the relationship between litter and soil microbial communities in different forest types. Additionally, your encouragement in data handling and result description will motivate us to continue our efforts. Once again, thank you for your valuable time and professional guidance. We believe that by making these revisions, we can further improve our research work, and look forward to your further feedback on the revised version.
Main comments:
Comments1: Abstract is too long and must be shortened, Instructions for authors says that ″The abstract should be a total of about 200 words maximum″.
Response1: We sincerely appreciate your valuable feedback. We have restructured and refined the abstract to succinctly encapsulate the key findings of this study.
Comments2: The map of the study areas should be added to the manuscript
Response2: Thank you very much for your valuable advice, I have added the map in Figure 1.
Comments3: Line 145 - ″litter″ should be ″Litter″
Response3: I apologize for the error and thank you for bringing it to my attention. It has been corrected.
Comments4: Lines 146-147, 345, 642 – the latin name of tree species should be in italic
Response4: I apologize for the error and thank you for bringing it to my attention. It has been corrected.
Comments5: Line 323 – ″asssembly″ should be ″assembly″
Response5: Thank you very much for your careful review and help us to point out the error. It has been corrected.
Comments6: Could you include to the end of conclusion 1-2 sentences on plans for further research?
Response6: Thank you very much for your valuable advice, which also prompts us to think about the direction for further in-depth research. I have added some notes on this.
Comments7: Chapter References must be formatted in accordance with the instructions for authors,
for example Journal Articles:
1. Author 1, A.B.; Author 2, C.D. Title of the article. Abbreviated Journal Name Year, Volume, page range.
Response7: Thank you very much for your valuable suggestions and careful review. We have rechecked the references in this article and made necessary revisions.
Comments8: Lines 536, 604 – CO2 - ″2″ should be as subindex
Response8: I apologize for the improper formatting and have checked and corrected the entire text.
Comments9: Line 630 – ″Abbreviated Journal Name Year, Volume, page range″ are missing
Response9: I apologize for missing this information and have checked and added it.
Reviewer 3 Report
Comments and Suggestions for Authors
The paper „Assembly characteristics and influencing factors of the soil microbial community in the typical forest of Funiu Mountain” is current and very well structurate.
The authors present a study case focused on three representative forest types in the Funiu Mountains—Larix gmelinii (LG), Quercus aliena var. acutiserrata (QA), and Quercus aliena var. acutiserrata + Pinus armandii (QAPA). They examined the diversity, composition, co-occurrence networks, assembly processes, and influencing factors of bacterial communities.
The authors' research has provided an essential insight into understanding the synergistic relationships between litter quality, soil characteristics and the microbial community in different forest ecosystems.
I propose to publish the paper in present form.
Author Response
The paper “Assembly characteristics and influencing factors of the soil microbial community in the typical forest of Funiu Mountain” is current and very well structurate.
The authors present a study case focused on three representative forest types in the Funiu Mountains—Larix gmelinii (LG), Quercus aliena var. acutiserrata (QA), and Quercus aliena var. acutiserrata + Pinus armandii (QAPA). They examined the diversity, composition, co-occurrence networks, assembly processes, and influencing factors of bacterial communities.
The authors' research has provided an essential insight into understanding the synergistic relationships between litter quality, soil characteristics and the microbial community in different forest ecosystems.
I propose to publish the paper in present form.
Response: First of all, we would like to express our heartfelt gratitude for your high evaluation and support of our research. We are delighted to learn that you believe our research provides important insights into the synergistic relationships between litter quality, soil characteristics, and microbial communities in different forest ecosystems. Your endorsement means a lot to us and has greatly boosted our confidence in continuing to explore this field further.
If we have the privilege of publishing this article, we will submit the final version according to the journal's requirements and ensure that all formatting and details meet publishing standards. If there are any questions or further clarifications needed during this process, we will promptly communicate with you. Again, we thank you for your valuable review and suggestions, which are invaluable assets to us.
Round 2
Reviewer 1 Report
Comments and Suggestions for Authors
Dear Editor-in-Chief,
Thank you for making available a revised version of the manuscript.
The authors worked hard on the corrections, and all the changes were made.
All the best.
Author Response
Comment: The authors worked hard on the corrections, and all the changes were made. All the best.
Response: Thank you so much for your recognition of our efforts. We are delighted to know that the corrections have met your expectations. Your comments and suggestions have been extremely valuable in improving our work. All the best to you as well.
Best regards,